# Direct Endoscopic Necrosectomy of a Recurrent Walled-Off Pancreatic Necrosis at High Risk for Severe Bleeding: A Hybrid Technique Using a Dedicated Device

**DOI:** 10.3390/diagnostics13142321

**Published:** 2023-07-10

**Authors:** Cecilia Binda, Chiara Coluccio, Antonio Vizzuso, Alessandro Sartini, Monica Sbrancia, Alessandro Cucchetti, Emanuela Giampalma, Stefano Fabbri, Giorgio Ercolani, Carlo Fabbri

**Affiliations:** 1Gastroenterology and Digestive Endoscopy Unit, Forlì-Cesena Hospitals, AUSL Romagna, 47121 Forlì, Italy; cecilia.binda@auslromagna.it (C.B.); alessandro.sartini@auslromagna.it (A.S.); monica.sbrancia@auslromagna.it (M.S.); stefano.fabbri@auslromagna.it (S.F.); carlo.fabbri@auslromagna.it (C.F.); 2Radiology Unit, Morgagni-Pierantoni Hospital, AUSL Romagna, 47121 Forlì, Italy; antonio.vizzuso@auslromagna.it (A.V.); emanuela.giampalma@auslromagna.it (E.G.); 3Department of Medical and Surgical Sciences, Sant’Orsola-Malpighi Hospital, University of Bologna, 40138 Bologna, Italy; alessandro.cucchett2@unibo.it (A.C.); giorgio.ercolani@auslromagna.it (G.E.); 4General and Oncologic Surgery, Morgagni-Pierantoni Hospital, AUSL Romagna, 47121 Forlì, Italy

**Keywords:** walled-off pancreatic necrosis, direct endoscopic necrosectomy, EUS-guided drainage, necrotizing pancreatitis, EndoRotor

## Abstract

Direct endoscopic necrosectomy (DEN) is a challenging procedure for the debridement of walled-off pancreatic necrosis (WOPN), which may be complicated by several adverse events, primarily bleeding which may require radiological embolization or even surgery. The lack of dedicated devices for this purpose largely affects the possibility of safely performing DEN which increases the risk of complications. We present the case of a 63 years-old man who underwent an endoscopic ultrasound (EUS)-guided drainage of a WOPN, and who was readmitted one month after stent removal with clinical, endoscopic, and radiological signs of infected necrosis involving the splenic artery. A second EUS-guided drainage was performed, with clear visualization of the arterial vessel in the midst of a large amount of solid necrosis. Due to the high risk of major bleeding during DEN, a hybrid procedure in the angiographic room was performed, in order to identify and avoid, under fluoroscopic control, the splenic artery during the entire procedure guide, which was successfully performed using the EndoRotor system. We hereby review the current literature regarding DEN using the EndoRotor system. The case reported, with a literature overview, may help the management of these patients affected by benign but life-threatening conditions which involve a multidisciplinary setting.

## 1. Introduction

In recent decades, especially after the introduction on the market of dedicated devices, Endoscopic ultrasound (EUS)-guided drainage of pancreatic fluid collections (PFCs), which represent a complication of acute pancreatitis (AP) (as described by the revised Atlanta classification) [1], has gradually and increasingly spread, becoming nowadays the first line treatment for such a life-threatening condition, as recommended by international guidelines [2,3]. In particular, in 20% of cases of AP a necrotic collection may develop and, among these, 30% can generate an infection of necrosis which, if untreated, is associated with a mortality rate of around 20–30% [4]. Direct endoscopic necrosectomy (DEN) is a challenging procedure first described in 2000 [5] which aims for the debridement of solid content in the context of walled-off pancreatic necrosis (WOPN) that could not be drained by the sole presence of a stent [6]. This procedure appears to be safe and effective for the treatment of infected collections after a failure of medical treatment, as indicated by a step-up approach, although it may be complicated by several adverse events (AEs), with an overall complication rate that could rank around 30% [7] and, above all, bleeding is one the most fearsome AEs [8] due to the difficulties of obtaining a rapid and adequate hemostasis, leading in some cases to radiological treatment with vascular embolization or even surgery [9].

Moreover, although it can be performed using a combination of different techniques, to date no specifically dedicated devices have been designed for DEN and several endoscopic accessories have been used (such as snares, baskets, forceps, etc.). Especially when there is a high amount of solid necrosis inside the collection, major vessels potentially involved can be hidden and the lack of dedicated devices largely affects the potential to safely perform DEN, increasing the risk of severe bleeding. Preventive coil embolization has been described for arterial bleeding prevention during DEN, although it could be affected by ischemic injuries [9]. During recent years, a novel automated mechanical endoscopic resection system called EndoRotor (Interscope, Inc., Whitinsville, MA, USA) has been introduced and applied for DEN, allowing for performing it under continuous endoscopic visualization, reducing both the time and number of DEN sessions [10]. This system consists of a flexible catheter, driven by an electronically directed console, which can pass through the working channel of the endoscope (of at least 3.2 mm in diameter) and, when inside the collection, has the power to suck, cut, and progressively remove solid debris before catching them in a trap [11]. Current literature encourages the use of this tool to achieve complete clearance of WOPN thus optimizing the procedural time and the rate of AEs, and indeed the risk of bleeding seems to decrease, favoring successful treatment despite the presence of vessels inside the collection [12].

## 2. Case Presentation

We present the case of a 63 years-old man who in January 2020 underwent an EUS-guided drainage of a WOPN using an electrocautery-enhanced lumen-apposing metal stent (EC-LAMS), which was removed after repeated DEN over a period of three weeks.

One month after stent removal, the patient was readmitted to our unit due to the onset of fever and abdominal pain. A computed tomography (CT)-scan showed the presence of WOPN recurrence with signs of infection (Figure 1). Therefore, an urgent EUS-guided drainage using a 20 × 10 mm EC-LAMS was performed. Both contrast enhanced EUS and CT revealed the presence of the splenic artery (SA) within the collection, which was not visible at the endoscopic view, hidden by the presence of a large amount of solid necrosis. Hence, because of the high risk of major bleeding during DEN, we decided to perform a hybrid procedure in the angiographic room. First, access through the common femoral artery was obtained, the SA was selectively cannulated, and a catheter was inserted in order to make the vessel visible under fluoroscopy (Figure 2), although within limits due to the use of two-dimensional imaging. DEN was then performed using the automated mechanical endoscopic resection device called the EndoRotor system (Interscope, Inc., Whitinsville, MA, USA) (Figure 3); it was used from high (1700 revolution per minutes (rpm)) to low (1000 rpm) rotating speeds with a progressive increase of suction from 40 L/min to 60 L/min. The entire procedure was performed using both endoscopic and fluoroscopic control, allowing us to constantly identify and avoid the SA. DEN was completed in four hours without complications, avoiding a prophylactic embolization of the SA. At the end of the procedure, the EC-LAMS was removed and a synthetic hemostatic agent was applied for the prevention of delayed bleeding.

No post-procedural complications were observed, and at a 6th month follow-up, no evidence of recurrence was highlighted (Figure 4).

## 3. Discussion

The Endorotor System has been introduced in the market in 2016, after demonstration of its technical properties and therapeutic potential in live animals [13], and it was firstly designed to remove benign mucosal neoplastic tissue throughout the gastrointestinal tract [14,15]; after some years it was approved by the Food and Drug Administration for the removal of dead pancreatic tissue in December 2020 [16].

To date, eight studies have been published regarding DEN using the EndoRotor system, which include three case reports [12,17,18], one of which occurred in a percutaneous setting [17], four case series [10,19,20,21], and one large multicenter prospective trial [22] (summary of the studies in Table 1).

The first report of its use for the debridement of necrotic tissue dates to 2018, when Van der Wiel et al. described their first experience in two patients with acute necrotizing pancreatitis with signs of infected necrosis, successfully treated with this device and without AEs [10].

In 2020, the same author [19] published about a series of 12 patients with infected necrotic post-pancreatitis collections, and a total of 27 procedures of DEN using the EndoRotor system. Of these, nine patients were treated de novo while three patients had already undergone unsuccessful endoscopic necrosectomy procedures using conventional tools. An average of two procedures (range 1–7) per patient was required to achieve the complete removal of necrotic tissue and no procedure-related AEs occurred.

In 2020, the first Italian case of a large WOPN treated with Endorotor has been reported [12], in which a major vessel (superior mesenteric artery) was observed inside the collection (similarly to our case) so its prophylactic radiological embolization was aborted. After two DEN sessions without any AEs, only minimal debris remained in the area proximal to the artery and, at a follow up three weeks later, LAMS was removed and the patient remained in good clinical condition thereafter. After a while, the same group confirmed this positive trend in a case series of four patients [20], and reported for the first time the use of an Endorotor catheter in the new available size of 6 mm for the treatment of two collections, assuming a potential decrease in the number of DEN sessions required [21].

Among the reports, there is also one about the use of this device in a percutaneous setting [17] for a young man affected by a large infected WOPN located in the left and right retroperitoneum, considered unsuitable for transgastric interventions. After three months from the initial percutaneous drainage, the necrotic tissue in the left retroperitoneum was still in place, so the cutaneous fistulous tract was dilated to allow for the insertion of a flexible endoscope aimed to perform necrosectomy with Endorotor; no AEs were observed, except for the persistence of the cutaneous fistula for less than a week.

In 2021, a large multicenter prospective study was published [22], reporting the results of 30 patients and 64 DEN using the EndoRotor system. The median volume of the necrotic collection, assessed with CT or with EUS was 395 cm^3^, and the median percentage of necrotic debris within WOPN was 75%. After EUS-guided drainage with stent placement (see Table 1), the median number of procedures needed per patient was 1.5 and the overall median percent necrotic debris removed per procedure was 66%. At the 21-day post-necrosectomy follow-up visit, imaging confirmed that at least 70% removal of the necrotic debris was achieved in twenty-nine out of thirty patients (97%). In this study, some patients had severe post-procedural AEs (nine out of thirty patients), but only three out of these nine were adjudicated as possibly related to the DEN procedure: two gastrointestinal bleeding and one pneumoperitoneum. The two cases of bleeding were successfully endoscopically treated, without further relevant sequelae. The patient who experienced pneumoperitoneum (that was attributed to the torquing of the scope within the LAMS, leading to extravasation of free air into the peritoneal cavity) developed multisystem organ failure resulting in death 10 days later, but the event was adjudicated as unrelated to the EndoRotor device or DEN procedure, but rather to the severity of the underlying disease in a frail patient with multiple comorbidities. With regard to nonserious AEs, two out of eleven were (possibly) related to DEN: stent dislocation and anemia, solved after a single blood transfusion.

It should be stated that the definition of a percentage of necrosis debridment as a successful procedure was inconsistent between studies, as clinical remission of the septic state was the desired outcome, but a threshold of 80% debridment, mostly quantified with contrast enhanced CT, was achieved in the vast majority of cases, as we can see from Table 1. From the available data, the median number of procedures required to achieve complete removal of pancreatic necrosis using the Endorotor system was two per patient, which appears quite affordable compared to the standard technique, historically considered as a demanding and time-consuming procedure, requiring multiple sessions with subsequent prolonged hospital stays and increased costs [23]. This optimization of the procedure can also be emphasized after the introduction of the new 6-mm EndoRotor catheter [21], which has a 4.4-times larger cutting window and a 2.5-times larger inner lumen compared to the traditional one, allowing for an 8-times greater throughput and possibly quicker and more impactful DEN.

In the only prospective study available [22], the reported mean total procedure time was 117 min (mean EndoRotor time 71 min) and 10 days as a median of discharge time after the index DEN, compared with the 14 to 29 days reported in previous studies using conventional DEN techniques [24,25,26]. Based on these results, we can assume that DEN with the Endorotor system could be lead to shorter hospitalization time, but we still lack sufficient data to quantify its cost-effectiveness and to specify its overall morbidity, despite the low rates of AEs reported overall (mainly not specifically related to the device).

The main concerns regarding AEs are represented by major bleeding and perforation due to the involvement of crucial anatomical structures within or very close to the collection, as well as stent misdeployment or migration. In order to minimize these risks, an accurate pre-procedural study of the anatomical limits of the collection must be conducted, combining radiologic and endoscopic techniques. The use of coil-embolization of vessels encased in the necrosis is reported to be an option, prior to the DEN, or as a rescue therapy in case of major intra-procedural bleedings [9].

To our knowledge, no other cases are reported in the literature using the dual endoscopic and radiological interventional approach described herein.

A recent retrospective study analyzing 97 consecutive patients who underwent DEN using LAMS, identified paracolic gutter extension, increasing Acute Physiology and Chronic Health Evaluation II score (APACHE II score), and >50% gland necrosis as negative predictors for success of the procedure [27]. We can speculate that it could be useful to consider these negative predictors in order to evaluate and select which cases may deserve a first approach of necrosectomy directly with EndoRotor.

## 4. Conclusions

The effectiveness of DEN is currently limited by the lack of dedicated tools. The EndoRotor system has been demonstrated to maximize the debridement of a large amount of necrosis. We reported for the first time a case of a large WOPN with an arterial vessel inside successfully managed by a dual endoscopic and radiological interventional approach, minimizing the possibility of bleeding without the need for prophylactic embolization of the SA, thus avoiding an additional invasive procedure. In our opinion, the description of our challenging case, together with a literature overview on this topic, may help in real-life for the management and clinical decisions of this group of patients affected by benign but nevertheless life-threatening conditions which involve a multidisciplinary setting, particularly gastroenterologists/endoscopists, radiologists, and surgeons, who must work together in order to reach the optimal clinical outcome. Further investigations are needed to clearly define the safety and effectiveness of this intervention, as well as the spectrum of indications in which the use of dedicated but costly devices can add an edge.

## Figures and Tables

**Figure 1 diagnostics-13-02321-f001:**
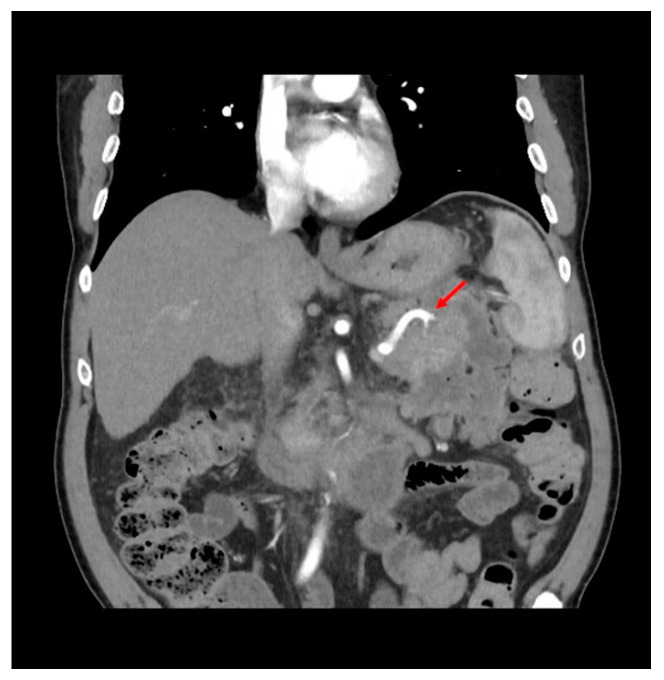
View of recurrent collection at pre-operative CT-scan. Red arrow points to the splenic artery inside the collection.

**Figure 2 diagnostics-13-02321-f002:**
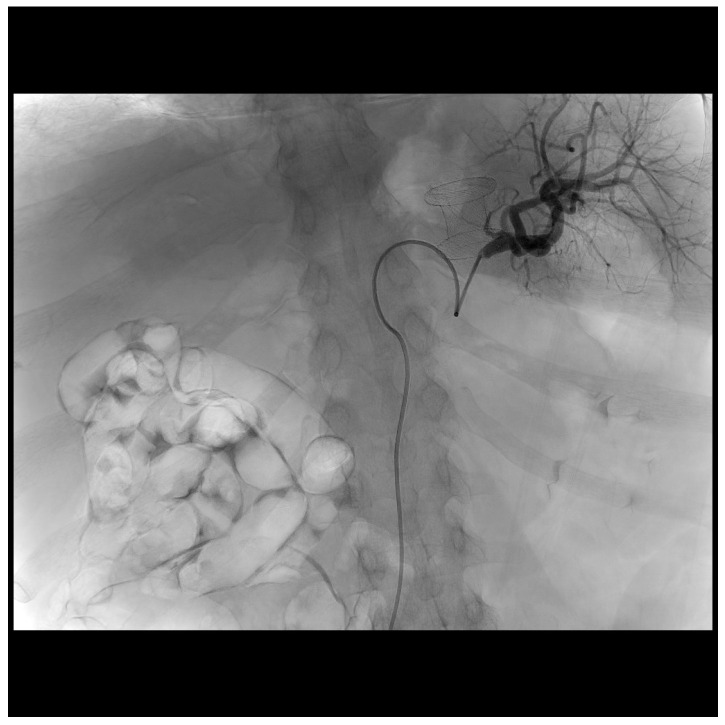
Fluoroscopic view of selective cannulation of splenic artery with a catheter inserted through the common femoral artery.

**Figure 3 diagnostics-13-02321-f003:**
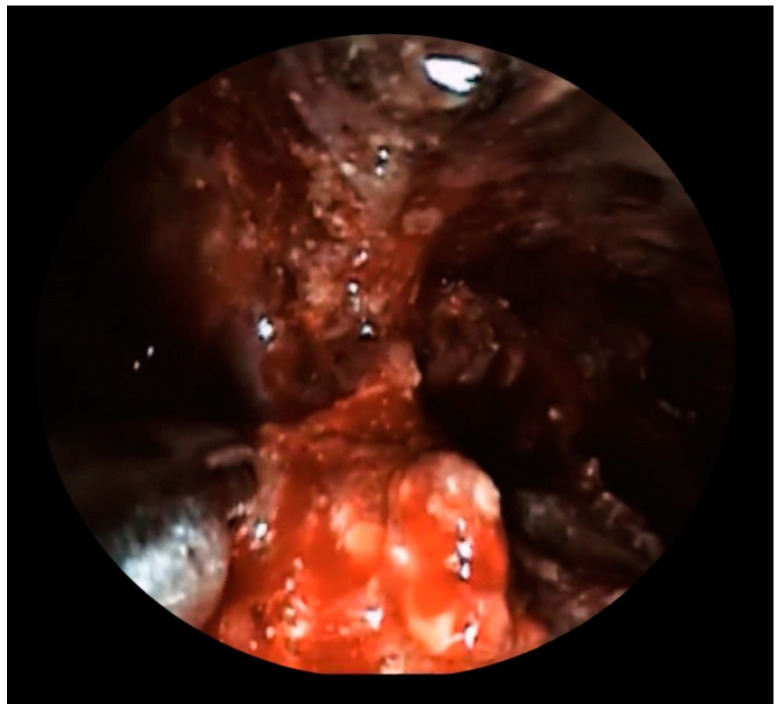
EndoRotor catheter coming out from the operative channel of the endoscope into the necrotic tissue of the collection.

**Figure 4 diagnostics-13-02321-f004:**
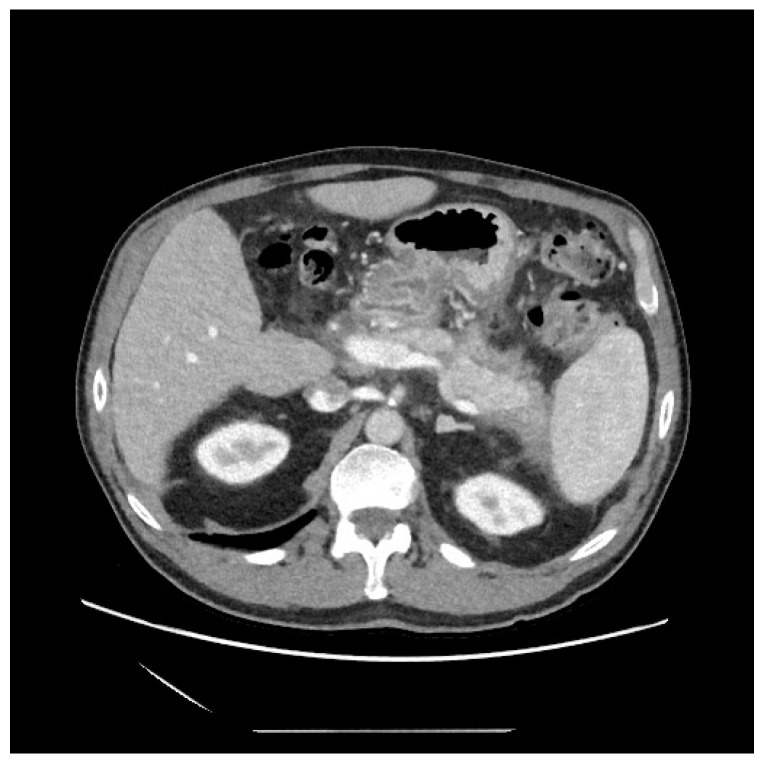
CT-scan at 6th month follow-up, showing the complete resolution of necrosis.

**Table 1 diagnostics-13-02321-t001:** Studies related to the use of the EndoRotor system during direct endoscopic necrosectomy (DEN) of walled-off pancreatic necrosis (WOPN).

Articles, Year of Publication	Patients (*n*)	Intra- Procedural AEs	Late AEs (>24 h)	Clearance of >80% Necrosis	Failure of DEN Using EndoRotor	LAMS	Plastic Stents	SEMS	Previous Unsuccessful DEN (*n*)
Van der Wiel et al., 2018 [10]	2	0/2 (0%)	0/2 (0%)	2/2 (100%)	0/2 (0%)	2	0	0	2
Bazarbashi et al., 2019 [18]	1	0/1 (0%)	0/1 (0%)	1/1 (100%)	0/1 (0%)	1	0	0	0
Rizzatti et al., 2020 [12]	1	0/1 (0%)	0/1 (0%)	1/1 (100%)	0/1 (0%)	1	0	0	0
Van der Wiel et al., 2020 [19]	12	0/12 (0%)	0/12 (0%)	11/12 (92%)	1/12 (8%)	4	8	0	3
Rizzatti et al., 2020 [20]	4	0/4 (0%)	0/4 (0%)	4/4 (100%)	0/4 (0%)	4	0	0	1
Stassen et al., 2021 [22]	30	0/30 (0%)	3 (10%)	29/30 (97%)	1/30 (3%)	23	2	5	Not specified
Zeuner et al., 2022 [17]	1	0/1 (0%)	0/1 (0%)	1/1 (100%)	0/1 (0%)	0	1	0	0
Rizzatti et al., 2023 [21]	2	0/2 (0%)	0/2 (0%)	2/2 (100%)	0/2 (0%)	2	0	0	0

Values are *n* (%) unless otherwise defined; AEs: Adverse Events. LAMS: Lumen-Apposing Metal Stent; SEMS: Self-Expandable Metal Stent.

## Data Availability

Data available in a publicly accessible repository.

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
