# Peer review of "Direct Endoscopic Necrosectomy of a Recurrent Walled-Off Pancreatic Necrosis at High Risk for Severe Bleeding: A Hybrid Technique Using a Dedicated Device"

_diagnostics, 2023, doi:10.3390/diagnostics13142321_

Round 1

Reviewer 1 Report

thank you for allowing me to review this original clinical case combining an endoscopic and radiological interventional approach. the illustrations are of high quality. the literature review is redundant in the text with the table provided and it would be useful to synthesize and summarize the ideas in the text. for example, how many cases have used the dual endoscopic and radiological interventional approach. similarly, how many procedures on average are needed to obtain effective treatment and what is the overall morbidity given the small number of cases reported. 

Van der Wiel appears twice in the table; are these the same patients? 

in the discussion, the authors cite a prospective series of 97 patients, but this does not appear in the table. why? 

lastly, i don't agree with the first sentence in the discussion, namely that the procedure is "safe and effective". i'd qualify this statement by pointing out the paucity of data in the literature. 

Author Response

Response to Reviewer 1 Comments

Point 1: thank you for allowing me to review this original clinical case combining an endoscopic and radiological interventional approach. the illustrations are of high quality. the literature review is redundant in the text with the table provided and it would be useful to synthesize and summarize the ideas in the text. for example, how many cases have used the dual endoscopic and radiological interventional approach. similarly, how many procedures on average are needed to obtain effective treatment and what is the overall morbidity given the small number of cases reported.

Response 1: Thank you for your positive comments and for the constructive suggestions, which I applied throughout the main text (specifying how many cases have been described, how many procedures are needed and overall morbidity, which I actually didn’t specify in number or percentage due to the paucity of data) synthesizing the contents.

Point 2: Van der Wiel appears twice in the table; are these the same patients?

Response 2: Thank you for this clarification. Van der Wiel appears twice in the table but they are two different studies: the first one is a case report dated 2018 (ref n° 11) while the second one is a case series dated 2020 (ref n°20).

Point 3: in the discussion, the authors cite a prospective series of 97 patients, but this does not appear in the table. why?

Response 3: Thank you for this clarification. I didn’t report the retrospective (rather than prospective as I wrote before, it was a mistake that I marked now in the text) series of 97 patients named in the discussion (from Zhai et al) because they didn’t use the Endorotor system in their series (while all the other studies reported in the table did). I highlighted this well published study only to hypothesize, as written in the text, that it could be useful to consider those negative predictors in order to evaluate and select which cases may deserve a first approach of necrosectomy directly with EndoRotor.

Point 4: lastly, i don't agree with the first sentence in the discussion, namely that the procedure is "safe and effective". i'd qualify this statement by pointing out the paucity of data in the literature.

Response 4: Thank you for this suggestion, I actually removed that sentence and pointed out the paucity of data in the conclusion.

Reviewer 2 Report

The authors described a direct endoscopic necrosectomy (DEN) of a recurrent walled-off necrosis at high risk of bleeding. They performed a hybrid procedure combining endoscopic and angiographic techniques to avoid the splenic artery injury. The case report was generally well-written. However, I have several concerns to improve the case report.

1. Why did walled-off necrosis recur after the first EUS-guided drainage? Did the patient have disconnected duct syndrome?

1. Figure 1: Please highlight the splenic artery with an arrow in the CT scan to improve the readers’ understanding.

2. How could the authors completely avoid the splenic artery using only two dimensional fluoroscopic imaging? Please show the technical details of how the authors avoided the vessel.

3. Please show endoscopic imaging during DEN.

4. Why did the authors remove LAMS immediately after the procedure without follow-up CT? Generally, LAMS is left in situ after DEN for a short period.

5. The contents regarding EndoRotor between ‘Review of the literature’ section and the Discussion section overlap. Please shorten them.

Minor editing of English language required.

Author Response

Response to Reviewer 2 Comments

The authors described a direct endoscopic necrosectomy (DEN) of a recurrent walled-off necrosis at high risk of bleeding. They performed a hybrid procedure combining endoscopic and angiographic techniques to avoid the splenic artery injury. The case report was generally well-written. However, I have several concerns to improve the case report

Point 1: Why did walled-off necrosis recur after the first EUS-guided drainage? Did the patient have disconnected duct syndrome?

Response 1: Thank you for your positive comments and for the constructive suggestions. We actually didn’t make the diagnosis of disconnected duct syndrome at that time, despite we checked many times the imaging reports togheter with radiologists during multidisciplinary meetings. In the aftermath, we may hypothesize that or otherwise that the first step of our drainage was suboptimal.

Point 2: Figure 1: Please highlight the splenic artery with an arrow in the CT scan to improve the readers’ understanding.

Response 2: Thank you for your kind comment, I replaced the figure 1 adding a red arrow pointing the splenic artery.

Point 3: How could the authors completely avoid the splenic artery using only two dimensional fluoroscopic imaging? Please show the technical details of how the authors avoided the vessel.

Response 3: Thank you for your comment. I actually now specified at line 98 (highlighted version) the use of a two-dimensional fluoroscopic imaging, which represents a limit, nonetheless the presence of the catheter during the entire procedure gave us the chance to inject contrast inside the vessel in every moment the procedure, especially when Endorotor catheter was very close to the vessel, helping us to constantly identify and avoid it.

Point 4: Please show endoscopic imaging during DEN.

Response 4: Thank you for this suggestion. I replaced the Figure 3 with a picture of Endorotor catheter inside the collection during DEN, to improve the readers’ understanding.  

Point 5: Why did the authors remove LAMS immediately after the procedure without follow-up CT? Generally, LAMS is left in situ after DEN for a short period.

Response 5: Thank you for giving me the opportunity to clarify this point. We actually didn’t remove LAMS immediately but, as now better specified at line 88 (highlighted version), it was removed after repeated DEN over a period of three weeks and a CT-scan which demonstrated a trend towards the resolution of the collection. Given the constant clinical and laboratoristic improvement, together with a pushing from the patient himself to discharge for personal reasons, we decided to remove the LAMS and start a close follow-up.

Point 6: The contents regarding EndoRotor between ‘Review of the literature’ section and the Discussion section overlap. Please shorten them.

Response 6: Thank you for the advice, I combined and synthesized the discussion and review contents as suggested.

Point 7: Minor editing of English language required.

Response 7: Thank you for the advice, I made some minor editing of englisg language.

Round 2

Reviewer 2 Report

The paper is revised properly according to the reviews' recommendation.

The quality of English is good.